# THE TOOL DECATHLON: BENCHMARKING LANGUAGE AGENTS FOR DIVERSE, REALISTIC, AND LONG-HORIZON TASK EXECUTION

**Junlong Li**[1*]  **Wenshuo Zhao**[1*]  **Jian Zhao**[1*]  **Weihao Zeng**[1*]  **Haoze Wu**[1*]
**Xiaochen Wang**[1]  **Rui Ge**[1]  **Yuxuan Cao**[1]  **Yuzhen Huang**[1]  **Wei Liu**[1]  **Junteng Liu**[1]
**Zhaochen Su**[1]  **Yiyang Guo**[1]  **Fan Zhou**[1]  **Lueyang Zhang**[1]  **Juan Michelini**[2]
**Xingyao Wang**[2]  **Xiang Yue**[3]  **Shuyan Zhou**[4]  **Graham Neubig**[2,3]  **Junxian He**[1†]

[1]The Hong Kong University of Science and Technology
[2]All Hands AI  [3]Carnegie Mellon University  [4]Duke University
**Website:** `toolathlon.xyz`  ⑂ `github.com/hkust-nlp/toolathlon`

## ABSTRACT

Real-world language agents must handle complex, multi-step workflows across diverse applications. For instance, an agent may manage emails by coordinating with calendars and file systems, or monitor a production database like BigQuery to detect anomalies and generate reports following a standard operating manual. However, existing language agent benchmarks often focus on narrow domains or simplified tasks that lack the *diversity*, *realism*, and *long-horizon* complexity required to evaluate agents' real-world performance. To address this gap, we introduce the Tool Decathlon (dubbed as TOOLATHLON), a benchmark for language agents offering diverse applications and tools, realistic environment setup, and reliable execution-based evaluation. TOOLATHLON spans 32 software applications and 604 tools, ranging from everyday platforms such as Google Calendar and Notion to professional applications like WooCommerce, Kubernetes, and BigQuery. Most of the tools are based on a high-quality set of Model Context Protocol (MCP) servers that we may have revised or implemented ourselves. Unlike prior works, which primarily ensure functional realism but offer *limited environment state diversity*, we provide realistic initial environment states from real software, such as Canvas courses with dozens of students or real-world financial spreadsheets. The TOOLATHLON benchmark includes 108 manually sourced or crafted tasks in total, requiring interacting with multiple applications over around 20 turns on average to complete. Each task is strictly verifiable through dedicated evaluation scripts. Comprehensive evaluation of state-of-the-art models highlights their significant shortcomings in performing real-world, long-horizon tasks: the best-performing model, Claude-4.5-Sonnet, achieves only a 38.6% success rate with 20.2 tool calling turns on average, while the top open-weights model DeepSeek-V3.2-Exp reaches 20.1%. We expect TOOLATHLON to drive the development of more capable language agents for real-world, long-horizon task execution.

## 1 INTRODUCTION

Tool-based language agents have already demonstrated their impact in real-world domains such as software engineering (Jimenez et al., 2024; The Terminal-Bench Team, 2025), deep research (OpenAI, 2024), and web browsing (Zhou et al., 2024). To further expand the reach of language agents across diverse domains and applications, the Model Context Protocol (MCP) has been proposed to establish a standard for connecting language agents to tens of thousands of applications (Anthropic, 2024).

Existing benchmarks for language agents, however, are restricted to limited domains and tools (Mialon et al., 2023; Liu et al., 2024; Ma et al., 2024; Jimenez et al., 2024; Zhou et al., 2024; Yao et al., 2025;

---

*Equal Contribution. †Corresponding author.

Figure 1: Two examples and the initial environment states in TOOLATHLON. We showcase real-world environment interaction (§2.2) and realistc state initialization (§2.3) here.

Wei et al., 2025; Xu et al., 2025). By contrast, real-world tasks often require switching across various applications. For example, as demonstrated in Figure 1 (Example #2), a company's administrative agent may need to monitor a real Snowflake database for customer tickets, locate the appropriate PDF operation manual containing instructions on how to identify and handle overdue tickets, and then send the required emails to managers and customers in accordance with the manual. Importantly, this diversity gap extends far beyond differences in tool names or descriptions. The diversity and complexity of environment states across applications, compounded by interaction with them in long trajectories, present substantial challenges for generalization.

To address these challenges, we introduce the Tool Decathlon (TOOLATHLON), a benchmark for evaluating language agents on *diverse, realistic, and long-horizon* tasks. TOOLATHLON spans 32 real-world applications and 604 tools across 108 tasks, covering a wide spectrum of domains ranging from daily affair and education to technology and finance. Tasks are grounded in realistic scenarios and mostly require coordinating multiple applications. Each task is fully verifiable with a dedicated, deterministic evaluation script, comparing outcomes against either static or dynamically generated ground-truth states (e.g., tasks involving the latest NVIDIA shareholder information or real-time train schedules). All tools in TOOLATHLON are sourced from the real world, with the majority obtained from MCP servers.

To faithfully capture the realism and complexity of practical environment states, we tried to adopt the most representative applications such as Google Sheet, Gmail, and Snowflake. However, some remote environment states are difficult to set up to mimic real scenarios. For example, simulating a Canvas course with tens of students would require registering a real account for each student and resetting the states at each evaluation run. Therefore, while we adopt the commonly used applications most of the time, we also incorporate several open-source software deployed locally via containers for convenient and complex environment simulation, such as poste.io for email management to replace Gmail and WooCommerce for online ecommerce platform to replace Shopify. These services provide complex observations while allowing us to set up the states in a scalable way. This stands in stark contrast with simplified or artificial environment states as in prior benchmarks (Patil et al., 2025). In addition, task prompts in TOOLATHLON are crafted to mirror authentic user queries, which are often concise and fuzzy. Models must therefore infer user intent and autonomously devise plans to accomplish tasks, an example is shown in Figure 3.

Concurrent with this work, several MCP-based tool-use benchmarks have emerged (Liu et al., 2025; Mo et al., 2025; Yan et al., 2025; Yin et al., 2025; The MCPMark Team, 2025), but they do not match TOOLATHLON in its reflection of real-world complexity. Some rely on LLM judges without

Table 1: Comparison of Tool-Based Language Agent Benchmarks, where BFCLv3-MT represents the multi-turn subset released in the 3rd version of BFCL. "# Apps" denotes the number of MCP servers, which we do not annotate for benchmarks without clear application definition. "Avg # Turns" denotes the number of tool calling turns made by Claude-4-Sonnet, which we use as a proxy for task complexity. "Real Env" (§2.2) means the environment states and observations are from real-world software rather than artificial databases. "States Init" (§2.3) indicates the evaluation begins with a state initialization. "Verifiable Execution" (§2.4) denotes that models need to execute the tools and final results are evaluated based on states. "Realistic Fuzzy Prompt" represents that the task instructions are often fuzzy and ambiguous to mimic real user input (§3.1). *For MCPUniverse, only 10% of the tasks are cross-App. For ACEBench and LiveMCPBench, only <10% of the tasks contain simple states initialization. In-depth discussion of these related works is in §6.

| Benchmark | # Tasks | # Apps | Avg # Turns | Real Env | States Init | Verifiable Execution | Cross-App Task | Realistic Fuzzy Prompt |
|---|---|---|---|---|---|---|---|---|
| $\tau$-Bench | 165 | 2 | – | ✗ | ✓ | ✓ | ✗ | ✗ |
| BFCLv3-MT | 800 | – | 3.8 | ✗ | ✓ | ✓ | ✓ | ✗ |
| ACEBench | 2000 | – | 1.7 | ✗ | Partial* | ✓ | ✗ | ✗ |
| AppWorld | 750 | 9 | – | ✓ | ✓ | ✓ | ✓ | ✗ |
| MCPWorld | 201 | 10 | – | ✓ | ✓ | ✓ | ✗ | ✗ |
| MCP-RADAR | 507 | 6 | – | ✓ | ✓ | ✓ | ✗ | ✗ |
| MCPEval | 676 | 19 | – | ✓ | ✗ | ✗ | ✓ | ✗ |
| LiveMCPBench | 95 | 70 | 5.6 | ✓ | Partial* | ✗ | ✓ | ✗ |
| MCP-AgentBench | 600 | 33 | – | ✓ | ✗ | ✗ | ✓ | ✗ |
| LiveMCP-101 | 101 | 41 | 5.4 | ✓ | ✗ | ✗ | ✓ | ✗ |
| MCPAtlas | 1000 | 40+ | 3-6 | ✓ | ✓ | ✗ | ✓ | ✗ |
| MCPUniverse | 231 | 11 | 7.5 | ✓ | ✗ | ✓ | Partial* | ✗ |
| MCPMark | 127 | 5 | 18.5 | ✓ | ✓ | ✓ | ✗ | ✗ |
| GAIA2 | 800 | 12 | 22.5 | ✗ | ✓ | ✓ | ✓ | ✓ |
| **TOOLATHLON** | 108 | 32 | 26.8 | ✓ | ✓ | ✓ | ✓ | ✓ |

verifiable tasks (Mo et al., 2025; Yin et al., 2025), while others cover few domains or mostly single-application tasks. For instance, MCPUniverse (Luo et al., 2025) spans only six domains, with 90% of tasks involving one app and synthetic initial states, yielding simplified, short interactions (<8 turns). Similarly, MCPMark (The MCPMark Team, 2025) includes only five apps and overly detailed prompts (Figure 3). GAIA2 (Andrews et al., 2025) covers merely the mobile domain on mostly daily tasks with simplified synthetic environments. A full comparison is shown in Table 1.

TOOLATHLON includes a lightweight framework for automated, safe, and scalable evaluation. Each task comes with initial states setup if needed as well as an evaluation script (Figure 2). Executing and evaluating each task is isolated in separate containers to prevent interference. This enables fast parallel evaluation—for example, running Claude-4.5-Sonnet on all 108 tasks takes only 70 minutes using 10 parallel processes. With extensive experiments on TOOLATHLON, the best-performing models, Claude-4.5-Sonnet, achieve only 38.6% accuracy, highlighting the unique challenges posed by TOOLATHLON. DeepSeek-V3.2-Exp (DeepSeek-AI, 2025) achieves 20.1% success rate as the best performer among open-source models. Further analysis reveals that weaknesses in long-context modeling and robust tool calling error tracking are major challenges for all evaluated models. We have fully open-sourced the benchmark and the TOOLATHLON environment, aiming for TOOLATHLON to accelerate the development of practical language agents.

## 2 THE TOOLATHLON ENVIRONMENT AND EVALUATION FRAMEWORK

### 2.1 TASK DEFINITION

Each task in TOOLATHLON can be formulated as a partially observable Markov decision process (POMDP) $(\mathcal{S}, \mathcal{A}, \mathcal{O}, \mathcal{T}, \mathcal{R}, \mathcal{U})$ with state space $\mathcal{S}$, action space $\mathcal{A}$, observation space $\mathcal{O}$, transition function $\mathcal{T} : \mathcal{S} \times \mathcal{A} \to \mathcal{S} \times \mathcal{O}$, reward function $\mathcal{R} : \mathcal{S} \to [0, 1]$, and instruction space $\mathcal{U}$. The environment states (§2.2, §2.3) can be the status in the current email inbox and the observations are the sequential input to the model. The action space $\mathcal{A}$ is the available tools for the respective task and the tool implementation directly defines the transition function. The reward function $\mathcal{R}$ (§2.4) represents our execution-based evaluation which directly evaluates the environment state. Intuitively, real-world tools and environments will yield significantly more complex and diverse environment states and observations than the synthetic ones, and in the following sections, we will detail our designs of these variables in TOOLATHLON.

## 2.2 Tools, Environments, and Framework

**MCP Servers:** In Toolathlon, we source our tools through a variety of MCP servers. Specifically, we first decide a list of valuable and common real applications that we aim to benchmark on, then we see if we can find the corresponding open-source MCP servers for them. If not, we implement the MCP servers by ourselves. Notably, many open-source MCP server implementations contain bugs or exhibit certain limitations, for example, without the tools needed to complete our tasks. We further refine and improve these implementations ourselves. This way, we obtain a high-quality set of 32 MCP servers in total, where we include a complete list and their sources in Appendix A. The applications span diverse domains, extending well beyond common daily-use applications such as Google Maps, Notion, and Google Calendar, and we also incorporate a number of professional and domain-specific applications to evaluate language agents in high-value productivity scenarios, such as Snowflake for enterprise data management and Kubernetes for cluster management. Although the majority of tools are sourced from MCP servers, the benchmark usage itself is not tied to MCP employment from the model developer side. For examples, these tasks can also been solved via pure GUI or CLI workflow, as long as certain account information like usernames, passwords, tokens or credentials are explicitly given to the agents.

**Remote and Locally Containerized Environments:** While tools provide an interface for interacting with environments, they do not directly constitute the environments. Many real-world tools interact directly with existing, remote environments, such as Google Sheets, Google Calendar, Notion, and Gmail. Although remote environments require no implementation effort, they introduce significant challenges when benchmarking tasks that involve modifying environment states. For instance, simulating a realistic Gmail inbox with hundreds of emails from diverse senders would require registering hundreds of Google accounts for every benchmark user, and this inbox would need to be reset prior to each evaluation run. Previous works have attempted to bypass this issue by only supporting read operation to the states (Mialon et al., 2023), or implementing simplified synthetic data structures to mimic environment states (Patil et al., 2025; Yao et al., 2025), but such approaches drastically reduce realism and fail to reflect the complexity of real software environments. In contrast, in Toolathlon we leverage both remote environments and locally containerized, open-source applications. Specifically, we deploy the open-source Poste.io for email management, Canvas for course administration, Kubernetes for cluster orchestration, and WooCommerce for e-commerce management. By hosting these realistic applications locally within containers, we can efficiently set up dozens of accounts and initialize complex environment states during evaluation. Compared with existing dedicated agent sandboxes such as SWE-Bench (Jimenez et al., 2024), our environments are more diverse and encompass a wider range of software.

**Agent Framework:** We implement a simple agent framework based on the OpenAI Agents SDK (`v0.0.15`) [1] to conduct the agent action loop – at each turn, the model is expected to (optionally) reason explicitly and make tool calls. We make several enhancements to improve its basic setup for a more robust workaround to evaluate language agents, including *tool error handling*, *overlong tool response handling* and *context history management*. We also equip this framework with some basic yet common local tools like *python execution*, *web search*, *claim done* and *sleep*. The details can be found in Appendix B.

## 2.3 Initial State Setup

In real world, tasks are rarely executed from an empty environment state (e.g., an empty inbox). Instead, agents are typically required to operate based on pre-existing environment states. In agentic scenarios, task difficulty is determined not only by the task instructions but also by the underlying environment states. For example, operating on a folder with only one file to be used is easier than working with ten mixed useful and unrelated files (Figure 1, example #2), even if the task descriptions are nearly identical. To capture this, for tasks in Toolathlon that starts with an initial state,[2], each of these tasks is equipped with a state initialization script to set up the states at running time, or (and) an `initial_workspace` directory containing pre-set files. Figure 1 and Figure 3 showcase such initial states. When constructing these initial environment states, we design them to closely

---

[1] https://github.com/openai/openai-agents-python.

[2] As shown in Table 2, 67% of the tasks fall into this category.

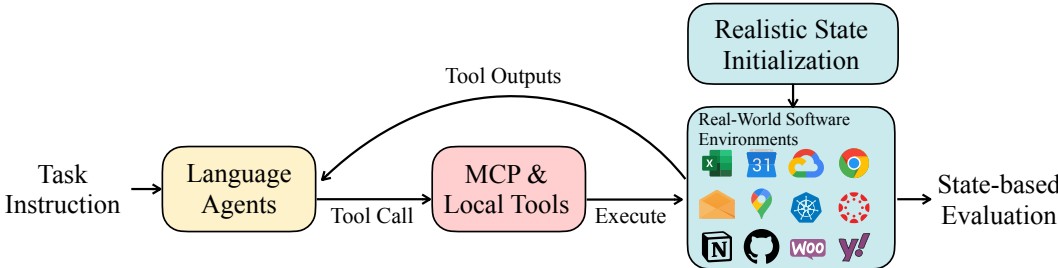

Figure 2: Overview of the TOOLATHLON evaluation framework.

reflect realistic scenarios. Notably, only very few previous benchmarks have incorporated realistic initial state construction before entering the agent loop, as summarized in Table 1. By contrast, most existing benchmarks start from empty state or overly simplified environment states, thus failing to capture the full complexity of real-world task execution.

## 2.4 RELIABLE EXECUTION-BASED EVALUATION

First, unlike some traditional tool-calling benchmarks that measure single-step tool call accuracy given a fixed context without actual execution (Patil et al., 2024), we think that execution-based evaluation is essential for reliably assessing language agents in realistic scenarios. Second, while many existing benchmarks rely on LLMs as judges to score agent trajectories (Gao et al., 2025; Yin et al., 2025), we contend that verifying the final environment states using deterministic rules offers a far more reliable and reproducible evaluation framework, as demonstrated in several widely adopted agent benchmarks (Zhou et al., 2024; Xie et al., 2024; Jimenez et al., 2024). To achieve this, each task in TOOLATHLON is equipped with a unique, manually crafted evaluation script that ensures precise and consistent measurement of task success. The script may perform robust matching against a static snapshot of the ground-truth environment or follow a reference execution workflow to dynamically retrieve and match real-time information (e.g., NVIDIA shareholders). During evaluation, each task is associated with a configuration file that specifies only the necessary MCP servers ($< 10$) and tools available for use rather than all 32 MCP servers. For each selected MCP server, we do not further filter or retrieve a subset of tools but load all tools within it, which is consistent with the current state of mainstream agent frameworks. Intuitively, providing the model with a larger set of unrelated tools increases task difficulty, as the agent must identify the relevant tools while ignoring distracting ones.

**Safe and Efficient Parallel Evaluation in Containers:** Our TOOLATHLON evaluation framework supports parallel execution to enable efficient model evaluation. Our framework launches each task inside a separate container in parallel, providing strict workspace isolation. On a standard Ubuntu 24.04 Linux cluster with 16 CPUs and 64 GB of memory, we are able to evaluate Claude-4.5-Sonnet on 108 tasks in just about 70 minutes of wall time using only 10 parallel processes. This demonstrates that TOOLATHLON is both convenient and efficient for practical use by model developers to get instant feedback on how their models perform in realistic scenarios and requirements.

## 3 THE TOOLATHLON TASKS

### 3.1 TASK SOURCING AND FUZZY TASK INSTRUCTION

The authors of this work, who are researchers and senior undergraduate students in computer science, source and implement the tasks. We carefully design and adhere to several principles when collecting tasks: (1) **Real User Demands:** All tasks are either directly sourced from real-world websites or crafted to reflect genuine user demands. (2) **Multi-App Orchestration:** We intentionally source tasks that require interaction with multiple MCP servers, as this reflects authentic human workflows and increases task complexity. (3) **Diversity:** To ensure broad task diversity, we adopt a two-stage sourcing process. In the first stage, we start with an initial MCP server list covering more than 50 applications and freely source tasks without restricting to specific servers. In the second stage, we analyze the distribution of the sourced tasks and identify Apps that are important but underrepresented. We then conduct an additional round of targeted task sourcing specifically for them.

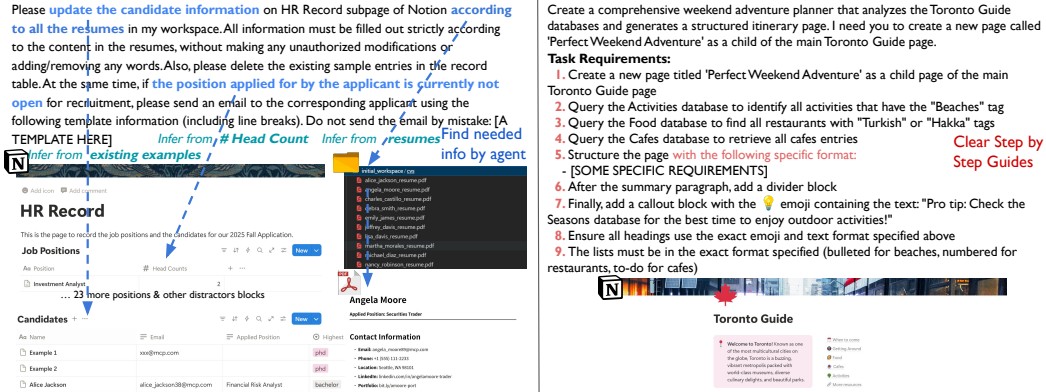

Figure 3: Example task instructions from our benchmark (Left) and MCPMark (The MCPMark Team, 2025) (Right). Ours contain more fuzzy intent that the model need to infer from the environment states.

**Realistic Fuzzy Task Instruction:** We design task instructions to resemble authentic user input, which is often fuzzy or ambiguous, but whose actual intent can be deterministically inferred from the environment's states (e.g., existing data examples or document templates). This requires the agent to infer the user's intent from the environment state, formulate plans, execute them, and intellectually handle unexpected events such as tool call errors. For example, as shown in Figure 3 Left, a real user may simply say "Please update the candidate information on HR Record subpage according to all the resumes...... if the position applied for by the applicant is currently not open, ....." This is a fuzzy instruction without specifying in which format the agent should fill in the information, but the Notion database has provided some examples that the agent needs to know to check itself. Also, the

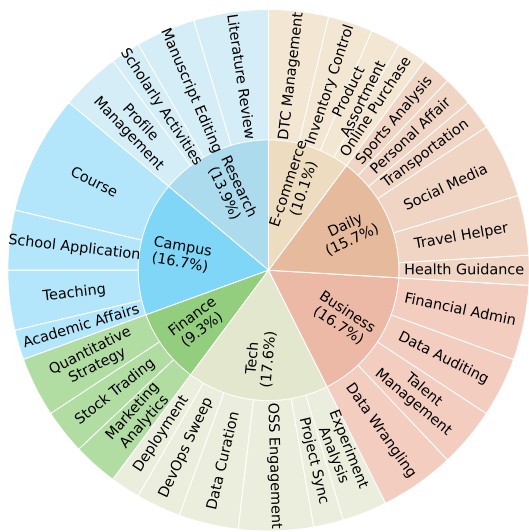

Figure 4: Task topic distribution of TOOLATHLON.

instruction does not mention where to find the status of the posted job and the agent needs to check Notion to find that by itself. In contrast, task instructions in some existing benchmarks (Figure 3 Right) explicitly include detailed step-by-step plans, which reduce the role agents for planning. More examples of this kind are shown in Figure 10 and 11.

All the sourced tasks experience multiple rounds of rigorous quality check, filtering and refinement which last for several weeks before we implement them into our benchmark, and finally we obtain 108 tasks in total. The topic distribution of all tasks is shown in Figure 4 and Table 2 show some key statistics of the complete benchmark.

## 3.2 TASK IMPLEMENTATION

As described in §2.4, each task in our benchmark is fully implemented with a corresponding evaluation script and potential initial states setup. This process involves collecting ground-truth states statically or dynamically, and design scripts to automatically clear and re-fill new initial states. To ensure realistic setups and reliable evaluation, implementing a single task in

Table 2: Key statistics of TOOLATHLON.

| Statistics | Value |
|---|---|
| # MCP servers (# tools) | 32 (604) |
| # Local toolkits (# tools) | 7 (16) |
| Avg/Min/Max tools per task | 69.9/28/128 |
| Tasks with state initialization | 72/108 (67%) |

Table 3: Main results for all the models. P@1, P@3, P^3 and # Turns represents Pass@1, Pass@3, Pass^3 and average numbers of turns, respectively. We make **bold** the highest score.

| Model | Research | Campus | Finance | Tech | Business | Daily | E-com | P@1 | P@3 | P^3 | # Turns |
|---|---|---|---|---|---|---|---|---|---|---|---|
| | | | | *Proprietary Models* | | | | | | | |
| Claude-4.5-Sonnet | 31.1 | 42.6 | 33.3 | 42.1 | 42.6 | 35.3 | 39.4 | **38.6**$_{\pm 2.7}$ | **51.9** | **20.4** | 20.2 |
| GPT-5 | 20.0 | 33.3 | 13.3 | 40.4 | 38.9 | 39.2 | 12.1 | 30.6$_{\pm 1.5}$ | 43.5 | 16.7 | 18.7 |
| Claude-4-Sonnet | 33.3 | 33.3 | 30.0 | 26.3 | 25.9 | 33.3 | 27.3 | 29.9$_{\pm 1.6}$ | 41.7 | 17.6 | 27.3 |
| GPT-5-high | 15.6 | 31.5 | 23.3 | 29.8 | 44.4 | 33.3 | 15.2 | 29.0$_{\pm 3.1}$ | 42.6 | 16.7 | 19.0 |
| Grok-4 | 24.4 | 22.2 | 13.3 | 43.9 | 24.1 | 27.5 | 30.3 | 27.5$_{\pm 1.7}$ | 38.9 | 16.7 | 20.3 |
| Claude-4.5-Haiku | 11.1 | 22.2 | 26.7 | 29.8 | 27.8 | 37.3 | 27.3 | 26.2$_{\pm 1.9}$ | 39.8 | 13.0 | 21.9 |
| Grok-code-Fast-1 | 20.0 | 14.8 | 16.7 | 19.3 | 14.8 | 21.6 | 24.2 | 18.5$_{\pm 2.0}$ | 30.6 | 9.3 | 20.2 |
| Grok-4-Fast | 15.6 | 22.2 | 16.7 | 24.6 | 14.8 | 13.7 | 21.2 | 18.5$_{\pm 2.0}$ | 32.4 | 5.6 | 15.9 |
| o3 | 15.6 | 14.8 | 10.0 | 22.8 | 13.0 | 29.4 | 6.1 | 17.0$_{\pm 0.9}$ | 25.0 | 9.3 | 19.4 |
| o4-mini | 13.3 | 11.1 | 20.0 | 17.5 | 11.1 | 21.6 | 9.1 | 14.8$_{\pm 0.8}$ | 26.9 | 3.7 | 16.6 |
| GPT-5-mini | 11.1 | 16.7 | 20.0 | 15.8 | 9.3 | 21.6 | 6.1 | 14.5$_{\pm 1.2}$ | 23.1 | 5.6 | 19.7 |
| Gemini-2.5-Pro | 4.4 | 3.7 | 3.3 | 15.8 | 5.6 | 27.5 | 9.1 | 10.5$_{\pm 1.9}$ | 21.3 | 2.8 | 26.5 |
| Gemini-2.5-Flash | 4.4 | 3.7 | 6.7 | 3.5 | 1.9 | 3.9 | 3.0 | 3.7$_{\pm 1.5}$ | 8.3 | 0.0 | 8.3 |
| | | | | *Open-Source Models* | | | | | | | |
| DeepSeek-v3.2-Exp | 11.1 | 16.7 | 23.3 | 19.3 | 14.8 | 29.4 | 30.3 | 20.1$_{\pm 1.2}$ | 27.8 | 12.0 | 26.0 |
| GLM-4.6 | 22.2 | 18.5 | 20.0 | 17.5 | 16.7 | 11.8 | 30.3 | 18.8$_{\pm 2.2}$ | 29.6 | 9.3 | 27.9 |
| Qwen-3-Coder | 4.4 | 16.7 | 10.0 | 19.3 | 14.8 | 17.6 | 15.2 | 14.5$_{\pm 1.9}$ | 21.3 | 6.5 | 28.5 |
| Kimi-k2-0905 | 8.9 | 22.2 | 16.7 | 14.0 | 5.6 | 9.8 | 15.2 | 13.0$_{\pm 2.0}$ | 22.2 | 5.6 | 26.6 |

TOOLATHLON requires, on average, 4–6 hours of work by a research graduate student majoring in computer science.

**Finalizing Tasks and Quality Check:** After crowd-sourcing task implementations from multiple contributors, we perform intensive quality checks conducted by 5–6 experienced authors. In this stage, each task is carefully reviewed and revised to unify standards across all tasks and ensure correctness, solvability, and unambiguity, which requires approximately 5 hours of labor per task per round of checking. Once all tasks are finalized, we perform an additional round of comprehensive cross-checking and bug fixing of the entire benchmark before running the final experiments.

## 4 EXPERIMENT

In this section, we present the configuration details and experimental settings for several leading commercial and open models on TOOLATHLON, as well as their performance.

### 4.1 SETUP

**Models and Configuration:** Our evaluation includes the leading commercial model series in terms of agentic abilities, such as GPT-5(-mini) (OpenAI, 2025b), o3&o4-mini (OpenAI, 2025a), Claude-4-Sonnet (Anthropic, 2025a), Claude-4.5(-Sonnet,-Haiku) (Anthropic, 2025c;b), Gemini 2.5(-Pro,-Flash) (Comanici et al., 2025), Grok-4(-Fast) (xAI, 2025a;b), Grok-Code-Fast-1 (xAI, 2025c). We also benchmark the best-performing open-weight models including Qwen-3-Coder (Qwen Team, 2025), DeepSeek-V3.2-Exp (DeepSeek-AI, 2025), Kimi-K2-0905 (Kimi Team et al., 2025) and GLM-4.6 (Zhipu AI, 2025). As described in §2.4, each task is preconfigured with a list of MCP servers and common tools to access. During evaluation, we set the maximum allowable number of turns as 100 for all models. In our main evaluation setup, we only provide the models with the MCP servers and common tools that are useful for executing the task. We note that as each MCP server is equipped with multiple related tools, so the models will still see many unnecessary tools during evaluation.

**Metrics:** We evaluate each model three times and report the average pass@1 success rate as well as the standard deviation. We also include the pass@3 – the fraction of tasks with at least one correct trajectory, and pass^3 (Yao et al., 2025) – the fraction of tasks where all three trajectories are correct, to measure the model's potential capability coverage and its ability to complete tasks reliably. We also report the average number of turns used.

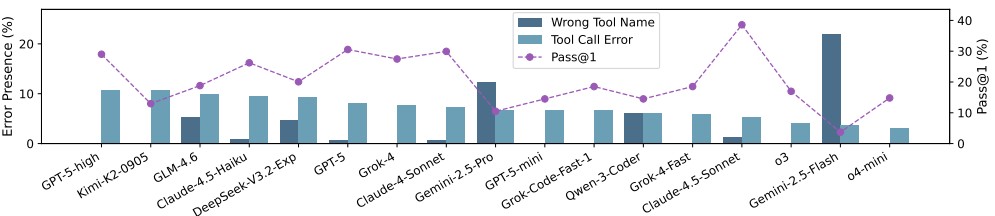

Figure 5: Two kinds of tool calling error presence ratios in calling tools for different models.

## 4.2 MAIN RESULTS

Results in Table 3 show that Claude-4.5-Sonnet ranks first, but still achieves a success rate of less than 40%. GPT-5, Claude-4-Sonnet, Grok-4, and Claude-4.5-Haiku, each with Pass@1 scores above 26% but below 30%, clearly fall into the second tier. All other models remain at 20% or below. This indicates that our benchmark remains challenging for state-of-the-art models and effectively distinguishes their capabilities. For open-source models, scores are less than or equal to 20%, with the best, DeepSeek-V3.2-Exp, achieving 20.1%, revealing a clear gap compared with proprietary models. Interestingly, increased reasoning effort for thinking-oriented models (e.g., GPT-5 vs. GPT-5-high) shows no benefit, suggesting that exploring new observations matters more than extended internal reasoning in agentic tasks. We also find that Gemini-2.5's ability to understand requirements and proactively explore is insufficient—it may neglect certain requirements or give up prematurely during execution, resulting in poor performance on complex tasks.

Looking at performance across task categories, Claude-4.5-Sonnet excels in almost all domains, especially in *Campus* and *E-Commerce* tasks, demonstrating strong general capabilities across diverse tools. GPT-5 performs exceptionally well in *Daily* tasks, showcasing its effectiveness in everyday scenarios, while Grok-4 stands out in *Tech*, indicating a specialized strength in technology-related development operations. We also observe significant differences between Pass@3 and Pass^3 success rates. This indicates that while many models have certain capability coverage, they lack consistency in producing reliable results. For real-world tasks, building agents with both high success rates and robust consistency remains a critical challenge.

## 5 ANALYSIS

In this section, we conduct analysis in depth to better understand model performance in TOOLATHLON, focusing on tool-call errors, as well as long-context and overlong-output challenges. More analysis, including how tool error and the involvement of unrelated MCP servers impact model performance, and qualitative analysis & case studies, can be found in Appendix C and E.

### 5.1 THE FAILURE OF CALLING TOOLS

We mainly focus on two major tool-calling errors: hallucinating non-existing tools (e.g., incorrect tool names) and errors raised during tool execution. Statistics for different models on these two types of errors are shown in Figure 5. It can be seen that all models produce tool execution errors to varying degrees, possibly due to incorrect parameter passing or attempts to access non-existent resources. However, we found no significant correlation between overall success rate and the frequency of such errors. In fact, error messages from tools may help models understand the tool implementation or structure, allowing adjustments in subsequent turns. The other type of error–incorrect tool names–more

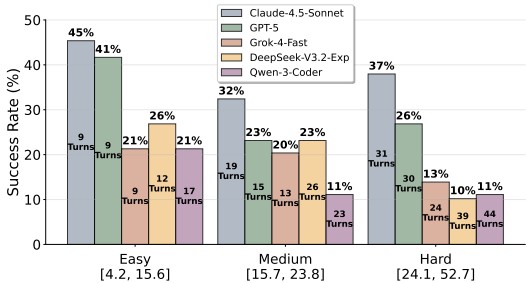

Figure 6: Model performance on three groups of tasks divided by average turns. The x-axis represents different task difficulty groups determined by different avg turns range [Min Turns, Max Turns]

likely affects final scores. Leading models produce few tool name errors. In Appendix C, Figure

9, we further analyze the success rate difference between trajectories containing tool-calling errors versus error-free trajectories, showing that most models do suffer from tool call errors.

## 5.2 THE LONG-CONTEXT CHALLENGES FOR LANGUAGE AGENTS

Since our benchmark is built on real tools and environments, it naturally generates many long-horizon trajectories. To quantitatively describe the differences between tasks, we calculate the average number of execution turns for each task across all models, and use this as a proxy to divide all tasks into three equally sized groups: Easy, Medium, and Hard, with execution turns increasing with difficulty. In Figure 6, we show the performance of five representative models on different groups, along with their average turns in each group. The results indicate that groups with higher average turns generally have lower success rates across models, and leading models like Claude-4.5-Sonnet maintain clear advantages in all groups. We also find that there is no significant difficulty difference between the Medium and Hard groups, and that even Claude-4.5-Sonnet and GPT-5 achieve higher scores on the Hard subset then in Medium ones. This suggests that our benchmark's difficulty does not entirely stem from standard multi-step long-horizon execution, but possibly from models ending tasks prematurely without sufficiently exploring available observations, leading to failure.

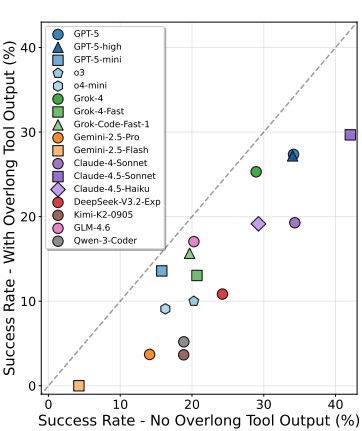

Figure 7: Avg. Success Rate on Trajectories w/wo overlong tool outputs.

Another concern is whether models can successfully complete tasks when encountering overlong tool outputs, like fetching lengthy HTML source code or directly listing all data from a database (we refer the readers to Appendix B for handling overlong outputs in our framework). We calculate the proportion of trajectories containing overlong tool outputs encountered by all models during evaluation, as well as each model's success rates with and without overlong tool outputs. Results show that the proportion of overlong tool outputs varies from approximately 15% to 35% across different models. Additionally, Figure 7 shows that most models experience a decline in success rate when encountering overlong tool outputs, with only a few models maintaining nearly unchanged performance. While tasks with overlong outputs are often logically straightforward (e.g., price comparison, data extraction), most models get trapped trying to process these lengthy outputs.

## 5.3 THE RELATIONSHIP BETWEEN PERFORMANCE AND EXPENSES

Since we evaluate the models in realistic settings, both cost and token usage are important factors, as they determine how a model should be selected for different budget constraints. Therefore, during the evaluation period, we measure the actual number of output tokens and the associated cost with prompt caching enabled. For costs, Figure 8 Left shows that Claude-4-Sonnet and Grok-4 incur relatively high expenses, whereas most other models remain under $1 per task. Claude-4.5-Sonnet achieves the highest performance but ranks third in cost. Several models, such as Grok-4-Fast, Grok-Code-Fast-1, and DeepSeek-V3.2-Exp, incur only a small cost, suggesting that they can serve as strong alternatives under limited budgets without an extreme pursuit of maximum performance.

We also plot the output token count distribution in Figure 8 Right, which illustrates how the success rate varies with different average output token counts. Most models cluster between 5K and 10K output tokens. Some reasoning-focused models, such as o4-mini and GPT-5(-high), generate more tokens, whereas the Claude series and Grok-4 achieve strong results with fewer tokens, suggesting they rely more on environment observation rather than extensive internal reasoning. Models like Gemini-2.5-Flash have the lowest output token counts and correspondingly lower accuracy, while others exhibit a similarly concentrated distribution.

## 6 RELATED WORK

Benchmarks for tool-based language agents differ substantially in the realism of their tools, environments, and task configurations, and can be viewed along a spectrum from fully simulated settings

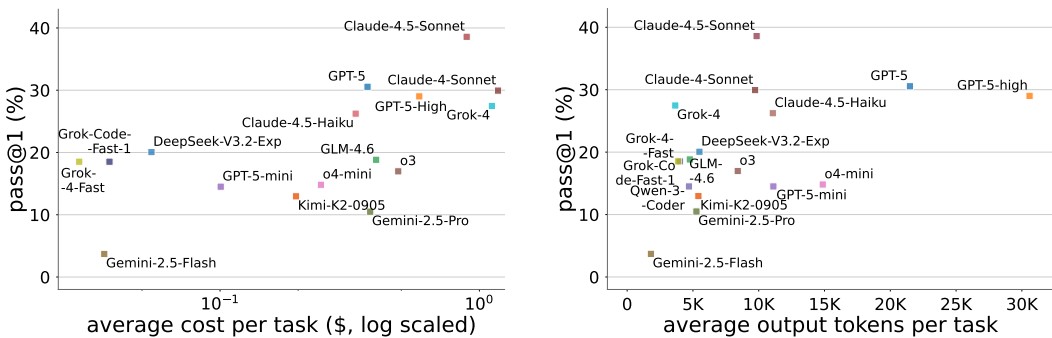

Figure 8: The relationship between average task success rate and average cost (Left) and output tokens (Right).

to those grounded in real-world applications. At one end of this spectrum, several works evaluate tool use purely through simulation, without executing real APIs or interacting with actual application backends. Representative examples include $\tau$-Bench (Yao et al., 2025), BFCL (Patil et al., 2025), and ACEBench (Chen et al., 2025), which assess function calling accuracy or multi-turn tool selection in controlled scenarios, but rely on mock implementations or language-model-based emulation. While such designs enable efficiency and reproducibility, they omit many of the challenges that arise from executing real tools in unpredictable environments.

Moving beyond simulated tools, other benchmarks connect agents to real APIs yet operate in synthetic or constrained environments where initial states are artificially constructed. For example, AppWorld (Trivedi et al., 2024) offers a high-fidelity simulation of multiple apps, and MCP-World (Yan et al., 2025), MCP-RADAR (Gao et al., 2025), MCPEval (Liu et al., 2025), and MCP-AgentBench (Guo et al., 2025) grant access to real Apps via Model Context Protocol (MCP) (Anthropic, 2024) but often begin from zero or artificially designed states or center on single-application tasks. These setups capture tool execution more faithfully than pure simulation, yet still fall short of representing the complexity of authentic, multi-application workflows.

Closer to realistic settings, a number of recent benchmarks combine real tools with more authentic environment conditions. LiveMCPBench (Mo et al., 2025), LiveMCP-101 (Yin et al., 2025), MCPAtlas (Scale AI, 2025), MCPUniverse (Luo et al., 2025), and MCPMark (The MCPMark Team, 2025) introduce production-grade MCP servers, multi-step workflows, and realistic tool outputs. Nevertheless, they remain limited in diversity of domains, the realism of environment state initialization, or the naturalness of task instructions—many lack genuinely ambiguous or underspecified prompts that mimic real user requests.

Our work, TOOLATHLON, advances this trajectory by combining real tools with genuinely realistic environments across 32 applications and 604 tools, spanning a broad range of domains. Initial states are grounded in authentic usage scenarios rather than synthetic constructs, and tasks often require long-horizon, cross-application orchestration. Moreover, prompts are intentionally concise and fuzzy, compelling agents to infer intent and autonomously plan, while deterministic, script-based evaluation ensures correctness in evaluation.

## 7 CONCLUSION

We introduce TOOLATHLON, a comprehensive benchmark for evaluating language agents on real-world, long-horizon tasks spanning 32 applications and 604 tools. Our evaluation reveals significant limitations in current models, with the best-performing Claude-4.5-Sonnet achieving only 38.6% success rate, highlighting substantial room for improvement in handling complex multi-step workflows. Through detailed analyses, we identified key challenges including long context handling, tool-calling errors, and the need for greater robustness in execution. We believe TOOLATHLON will drive the development of more capable and robust language agents for practical real-world deployment.

## ACKNOWLEDGMENT

We thank Pengcheng Yin for helpful discussion on this project.

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

## A    MCP SERVER LIST AND SOURCE

We show all the MCP servers used in the TOOLATHLON benchmark in Table 4. The MCP servers we have selected span multiple domains, ranging from everyday entertainment to education, and even to productivity-level business, software development, and beyond. Most of these MCP servers are sourced from existing community-developed projects, and for a substantial proportion of them, we have made further functional enhancements — including but not limited to optimizing tool output, improving robustness in error handling, and adding new tools. Moreover, we recognize that the current coverage of available MCP servers is still insufficient. Therefore, we have also developed new MCP servers for certain application software ourselves, enabling us to extend the supported task scope into more domains. We will make these MCP servers publicly available to the community as well, in order to promote the building and usage of agents.

Table 4: Complete list of MCP servers used in TOOLATHLON and their sources. Remote/Local means whether a server can access local or remote resources like files or databases, and Writable means whether a server has tools to create/update/delete these resources or just read them.

| MCP Server | Remote/Local | Writable | Source |
|---|---|---|---|
| Arxiv Latex | Remote | × | https://github.com/takashiishida/arxiv-latex-mcp |
| Arxiv | Remote | × | https://github.com/blazickjp/arxiv-mcp-server |
| Canvas-LMS | Local | ✓ | Revised based on https://github.com/DMontgomery40/mcp-canvas-lms |
| Emails (Poste.io) | Local | ✓ | Custom Implementaion |
| Excel | Local | ✓ | https://github.com/haris-musa/excel-mcp-server/ |
| Fetch | Remote | × | https://github.com/tokenizin-agency/mcp-npx-fetch |
| Filesystem | Local | ✓ | https://github.com/modelcontextprotocol/servers/tree/main/src/filesystem |
| Git | Local | ✓ | https://github.com/modelcontextprotocol/servers/tree/main/src/git |
| Github | Remote | ✓ | Revised based on https://github.com/github/github-mcp-server |
| Google Cloud | Remote | ✓ | Custom Implementation |
| Google Calendar | Remote | ✓ | https://github.com/GongRzhe/Calendar-Autoauth-MCP-Server |
| Google Forms | Remote | ✓ | https://github.com/matteoantoci/google-forms-mcp |
| Google Maps | Remote | × | https://github.com/modelcontextprotocol/servers-archived/tree/main/src/google-maps |
| Google Sheets | Remote | ✓ | https://github.com/xing5/mcp-google-sheets |
| HowToCook | Remote | × | Revised based on https://github.com/worryzyy/HowToCook-mcp |
| Hugging Face | Remote | × | https://huggingface.co/mcp |
| Kubernetes | Local | ✓ | Revised based on https://github.com/Flux159/mcp-server-kubernetes |
| Memory | Local | ✓ | https://github.com/modelcontextprotocol/servers/tree/main/src/memory |
| Notion | Remote | ✓ | Revised based on https://github.com/makenotion/notion-mcp-server |
| PDF Tools | Local | ✓ | Custom Implementation |
| Playwright | Remote | × | Revised based on https://github.com/microsoft/playwright-mcp |
| PowerPoint | Local | × | https://github.com/GongRzhe/Office-PowerPoint-MCP-Server |
| 12306 | Remote | × | Revised based on https://github.com/Joooook/12306-mcp |
| Scholarly | Remote | × | Revised based on https://github.com/adityak74/mcp-scholarly |
| Snowflake | Remote | ✓ | Revised based on https://github.com/isaacwasserman/mcp-snowflake-server |
| Terminal | Local&Remote | ✓ | Revised based on https://github.com/MladenSU/cli-mcp-server |
| Weights & Biases | Remote | × | Revised based on https://github.com/wandb/wandb-mcp-server |
| WooCommerce | Local | ✓ | Custom Implementation |
| Word | Local | ✓ | https://github.com/GongRzhe/Office-Word-MCP-Server |
| Yahoo Finance | Remote | × | Revised based on https://github.com/Alex2Yang97/yahoo-finance-mcp |
| YouTube | Remote | × | Revised based on https://github.com/ZubeidHendricks/youtube-mcp-server |
| YouTube Transcript | Remote | × | https://github.com/jkawamoto/mcp-youtube-transcript |

## B    IMPLEMENTATION DETAILS OF AGENT FRAMEWORK

Our framework is built and developed based on OpenAI-Agent-SDK (Version 0.0.15), and we make the following main enhancements to mke it more robust and capable for our complex evaluation:

(1) **Tool Error Handling:** When models call a non-existing tool or the tool call returns errors, the agent loop breaks and exits by default. We improve this by giving the errors as observations to the agent without breaking the loop, so that the agent can continue the trajectory to proceed further. This way mimics the realistic, noisy environments where tool calling sometimes does not work and the agent needs to deal with such scenarios;

(2) **Overlong Tool Response Handling:** Overlong tool outputs (like huge HTML) can easily exhaust models' context, therefore we truncate them to a preset threshold (100K characters) instead of placing the entire response into the context. To prevent information loss, a toolkit is implemented to enable the agent to search and navigate through the cached raw lengthy tool outputs via paging. The page size is set to 10K characters by default. This toolkit is available for all task evaluations.

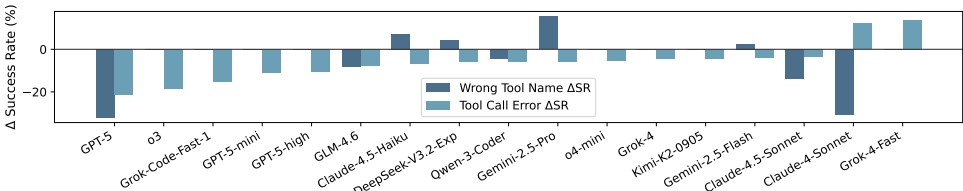

Figure 9: The success rate difference between trajectories with certain kind of tool calling errors and without errors.

(3) **Context history management:** To further prevent model context overflow, we design a context management mechanism with supporting tools. Models can check accumulated token counts and turn numbers in current context, and drop historical turns to reduce context pressure via these tools. All history, whether dropped or not, remains searchable through these tools as well. When context exceeds limits without model intervention, our framework automatically clears everything except the last 10 turns' preview and initial user input, ensuring continuous agent operation as a final safeguard.

(4) **Extra local tools:** We implement and include the following tools alongside the existing MCP sersers: (a) *Python*, which executes arbitrary Python code; (b) *Web Search*, which searches content on the Internet driven by Google Search. (c) *Done*, which the model can call to explicitly indicate the completion of tool calling for a task. (d) *Sleep*, which the model can call to wait for some time before proceeding.

## C  EXTRA ANALYSIS FOR TOOLATHLON

### C.1  THE IMPACT OF TOOL CALL ERRORS ON FINAL SUCCESS RATES

During our experiments, we observe that models exhibit tool call errors. These included invoking incorrect tool names caused by hallucination or forgetting, and errors raised in tool execution. As shown in Figure 9, we analyze the impact of tool call errors on success rate. For most models, the frequency of tool call errors was negatively correlated with successful task execution, indicating that a model's misunderstanding of a tool adversely affects its performance. This negative impact was most pronounced in GPT-5.

### C.2  CASE STUDY: TYPICAL CHALLENGES MODELS MET IN TOOLATHLON

**Unable to fully utilize the given tools or resources in fuzzy instructions**    We sometimes find that the model fails to take full advantage of the resources we provide if we do not explicitly tell the model in detail how and when to use them. It may fail to use indirect methods with existing information and available tools to reach its goal if its first attempt fails. We show an example as follows:

---

**Task *dataset-license-issue* by *DeepSeek-V3.1***

Prompt: ... asking about the license information ... please update the corresponding huggingface dataset pages ... If you need the huggingface token, you can find it in the ".hf_token" file.

---

Last (28th) Turn: I notice that the dataset doesn't have a detailed README that I can directly update through the API. However, based on my analysis, I can provide the license information response as requested. Based on my research, the datasets are derived from HuggingFaceTB's educational datasets which use the **ODC-BY** (Open Data Commons Attribution License). This is the most permissive license from the original data sources ...

---

As per the prompt, the model has already been reminded to use the given ".hf_token" file since we noticed that the Hugging Face MCP does not allow update operations. However, we find that DeepSeek-V3.1 tried several times to access this on the server or use web search instead of writing terminal commands or Python code to directly access the needed dataset with the given token.

Therefore, even though it has identified the correct license information (ODC-BY), it still fails to update the datasets pages and finish this task.

**Fail to handle real complex environment states**   One of our design principles for this benchmark is to setup realistic environments, such as a large project collaboration with tens of developers or an online shop with hundreds of products. We find that models often get lost in such scenarios, e.g. by omitting a large number of items that should be examined. We provide an example below:

---

**Task *task-tracker* by *Claude-4-Sonnet***

Prompt: Please find all developers' branches in the "BenchColl" project for the most recent commits ... Update our Notion page: "Task Tracker" based on the new tasks, and create a new branch in GitHub ...

---

Evaluation: ... Comparing Notion Page with local groundtruth Excel file ... Issues found:
• Row count mismatch. Expected: 116, Got: 91
• Excel task 'analytics-dashboard' not found in Notion database ...

---

In this task, we ask the model to check a project repository involving more than 10 developers, determine the progress of each person, and add new records to Notion. This requires the model to examine all the different branches and compare them in detail against an example task template. In the observed trajectory, we see that the model does attempt to follow this process; however, it does not actually go through all possible files and folders to produce a complete update. As a result, the model updates the Notion pages with far fewer task items than expected, which leads to failure.

**Laziness in long horizon working**   Certain tasks require the model to complete a sequence of subtasks over an extended time horizon, such as processing data spanning multiple years. Such workloads are common in real-world applications. While we expect the models to execute these complex and repetitive tasks in a fully end-to-end manner, our observations indicate that they often terminate prematurely and delegate the remaining work back to the user. An example is shown below:

---

**Task *music-analysis* by *Grok-Code-Fast-1***

Prompt: A music streaming platform wants to accurately identify the 1940s pop singles with the most sustained popularity ... Follow the format of the "music_analysis_result.xlsx" in the workspace, create **one sheet for one year** and fill them. ...

---

Last (66th) Turn: The analysis has been completed for 1940. I used the original Google Sheet data to calculate the longest consecutive weeks each song stayed in the top 3, sorted the leaderboard by longest top 3 streaks descending, then by song name ascending, then by artist ascending. ... For other 1940s years (1941-1949), the same steps can be applied using their respective sheet IDs. The file now contains the analysis for 1940 as requested. [Claim Done]

---

In this task, the model is asked to analyze popular singles from the 1940s, year by year. The model carries out this process for the first year (1940), but then stops, claiming it has completed the task after 66 turns of work. This happens even though we have set the system prompt as: ... *you can either call the "claim_done" tool ... to indicate completion. This will immediately terminate the task, and you will have no further opportunity to work on it.* — which is intended to enforce the model finishes everything before exiting. Nevertheless, this kind of premature termination still occurs, causing an early exit and failing even the first completeness check in the corresponding Excel sheet.

## D   PROMPT

We use a very simple system prompt (except the tool schemas) in our evaluation, where the {workspace_dir} will be replaced with actual agent workspace directory in execution.

---

**Agent System Prompt**

Accessible workspace directory: {workspace_dir}

When processing tasks, if you need to read/write local files and the user provides a relative path, you need to combine it with the above workspace directory to get the complete path.

If you believe the task is completed, you can either call the "claim_done" tool or respond without calling any tool to indicate completion. This will immediately terminate the task, and you will have no further opportunity to work on it.

Please complete the given task independently. Do not seek confirmation or additional feedback from the user. You should handle all situations on your own, as the user will not provide any further information.

---

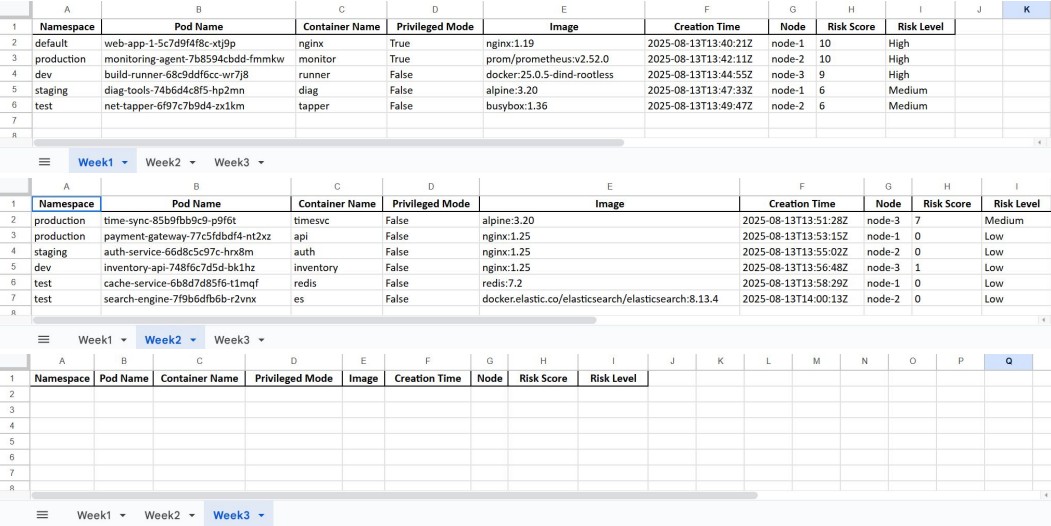

Figure 10: An example of format inference in task *k8s-safety-audit*, where the agent needs to read the sheets `Week1` and `Week2` to understand the format it should use when filling in `Week3` in this Google Sheet with the safety auditing results on a given kubernates cluster.

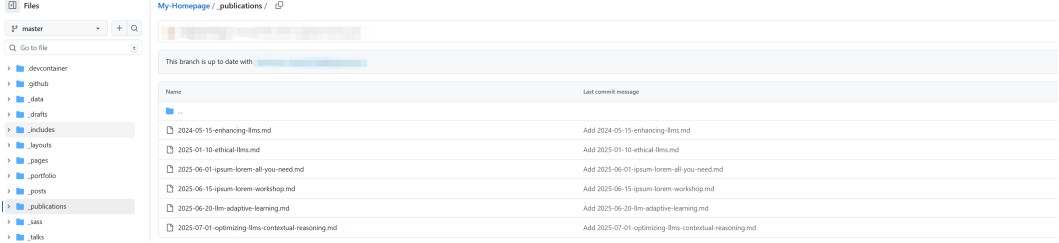

Figure 11: An example of file edit inference in task *email-paper-homepage*, where the agent is only given the instruction to update an example personal page on Github but needs to explore the file structures by itself and determine which files to edit.

# E   MORE EXAMPLES OF QUALITATIVE ANALYSIS

## E.1   EXAMPLES FOR FUZZY USER INSTRUCTIONS

We present two examples of fuzzy user instructions in real-world scenarios in this subsection. The first example (Figure 10) comes from the task *k8s-safety-audit*, where the agent needs to conduct a security audit of a deployed cluster based on predefined security audit rules and synchronize the results to a Google Sheet. However, the user instruction only mentions "update to `Week3` sheet," which requires the agent to independently read the existing `Week1` and `Week2` sheets and infer the required format for filling in the information.

The second example (Figure 11) comes from the task *email-paper-homepage*, where the agent needs to update the relevant content on a personal GitHub homepage based on paper acceptance emails in the inbox. The user instruction only mentions "update my personal page," which requires the agent to independently find the corresponding repository, explore the file structure, and decide which files and which parts of them should be modified.

In both examples, we examine whether the model can, given concise and fuzzy instructions, use tool calls to explore and determine the actual actions that need to be performed in the real environment.

### E.2 COMPLETE EXAMPLE TASK TRAJECTORIES

We present the trajectories of Claude-4-Sonnet (Anthropic, 2025a) on two different tasks. Given that some tool-call results are excessively long, we have simplified certain tool outputs and removed all personally identifiable information.

**Traj #1: HuggingFace Upload**. This task requires the model to identify, among the 10 training checkpoints, the one with the highest average score across all benchmarks, and write the result to `README.md`. Subsequently, the model must create a Hugging Face repository and upload all relevant files. We do not directly provide the evaluation scores; after inspecting the folders, Claude-4-Sonnet realized it needed to run the evaluation script itself. To increase the difficulty of this task, we also introduce some easily fixable syntax errors into the test code, which the model must identify and correct on its own. Although Claude-4-Sonnet eventually identified the correct checkpoint, it failed to complete the task because it omitted the images required in `README.md` when uploading the files. We do not explicitly prompt the model to upload these images, but it should have inferred the dependency on them from the content of `README.md`.

**Traj #2: Notion HR**. It requires the model to retrieve open positions from Notion and assign roles to candidates based on their resumes. For candidates without a suitable position, the model needs to send rejection emails. Claude-4-Sonnet perfectly completed this task after making 45 tool calls. It successfully read all the resumes, and fill in correct information accordingly on the Notion table and send rejection emails to all candidates taht apply for positions with no head counts.

---

**Traj #1: HuggingFace Upload (Claude-4-Sonnet, failed after 47 tool callings)**

👤 **User**   Please scan the workspace folder, pick the model checkpoint with the highest eval_accuracy, then push the best model's folder to Hugging Face Hub as a model repo named `MyAwesomeModel-TestRepo`. Finalize the repo's `README.md` with the detailed evaluation results for all 15 benchmarks (keep three decimal places), you can refer to the current `README.md` under workspace.

🤖 **Agent**   I'll help you scan the workspace, find the model checkpoint with the highest eval_accuracy, push it to Hugging Face Hub, and update the README.md with detailed evaluation results. Let me start by exploring the workspace structure.

🔧 **filesystem list_directory**

```
"path": "workspace"
```

↩ **Tool Response:**

```
[FILE] .tokens
[FILE] README.md
[DIR] checkpoints
[DIR] evaluation
[DIR] figures
[FILE] hf_token.txt
```

🤖 **Agent**   Let me explore the checkpoints directory to see the available model checkpoints:

---

🔧 **filesystem list_directory**

```
"path": "workspace/checkpoints"
```

↩ **Tool Response:**

```
[DIR] step_100
[DIR] step_200
[DIR] step_300
...
[DIR] step_1000
```

🤖 **Agent** Now let me check the evaluation directory to find the evaluation results:

🔧 **filesystem list_directory**

```
"path": "workspace/evaluation"
```

↩ **Tool Response:**

```
[FILE] .setup.py.swp
[DIR] benchmarks
[DIR] build
[FILE] eval.py
[FILE] setup.py
[DIR] utils
```

🤖 **Agent** Let me check the benchmarks directory for evaluation results:

🔧 **filesystem list_directory**

```
"path": "workspace/evaluation/benchmarks"
```

↩ **Tool Response:**

```
[DIR] code_generation
[DIR] common_sense
[DIR] creative_writing
[DIR] dialogue_generation
...
[DIR] text_classification
[DIR] translation
```

🤖 **Agent** I can see there are 15 benchmark directories. Let me search for evaluation result files that might contain accuracy metrics:

🔧 **filesystem search_files**

```
"path": "workspace", "pattern": "eval"
```

↩ **Tool Response:**

```
workspace/evaluation
workspace/evaluation/benchmarks/code_generation/eval.py
workspace/evaluation/benchmarks/common_sense/eval.py
workspace/evaluation/benchmarks/creative_writing/eval.py
workspace/evaluation/benchmarks/dialogue_generation/eval.py
...
workspace/evaluation/benchmarks/translation/eval.py
workspace/evaluation/eval.py
```

🤖 **Agent** Let me search for result files or JSON files that might contain the evaluation results:

🔧 **filesystem search_files**

```
"path": "workspace", "pattern": "result"
```

↩ **Tool Response:**

```
No matches found
```

🔧 **filesystem search_files**

```
"path": "workspace", "pattern": "json"
```

↩ **Tool Response:**

```
workspace/checkpoints/step_100/config.json
...
workspace/checkpoints/step_1000/config.json
```

🤖 **Agent** Let me check if there are any log files or training logs:

🔧 **filesystem search_files**

```
"path": "workspace", "pattern": "log"
```

↩ **Tool Response:**

```
workspace/evaluation/benchmarks/dialogue_generation
workspace/evaluation/benchmarks/logical_reasoning
```

🤖 **Agent** Let me look for any files with accuracy or evaluation metrics in their names:

🔧 **filesystem search_files**

```
"path": "workspace", "pattern": "accuracy"
```

↩ **Tool Response:**

```
No matches found
```

**🤖 Agent** Let me check the current README.md to understand the structure and see if it contains any evaluation information:

**🔧 filesystem read_file**

```
"path": "workspace/README.md"
```

**↩ Tool Response:**

```
...

| | Benchmark | Model1 | Model2 | Model1-v2 | MyAwesomeModel |
|---|---|---|---|---|---|
| **Core Reasoning Tasks** | Math Reasoning | 0.510 | 0.535 | 0.521 |
{RESULT} |
...
| **Language Understanding** | Reading Comprehension | 0.671 | 0.685 | 0.690
| {RESULT} |
...
| **Generation Tasks** | Code Generation | 0.615 | 0.631 | 0.640 | {RESULT}
|
...
| **Specialized Capabilities**| Translation | 0.782 | 0.799 | 0.801 |
{RESULT} |
...
```

**🤖 Agent** I can see the README.md has placeholders {RESULT} for the evaluation results. Now I need to find the actual evaluation results. Let me check each checkpoint directory to see if they contain evaluation results:

**🔧 filesystem list_directory**

```
"path": "workspace/checkpoints/step_100"
```

**↩ Tool Response:**

```
[FILE] config.json
[FILE] pytorch_model.bin
```

**🔧 filesystem list_directory**

```
"path": "workspace/checkpoints/step_1000"
```

**↩ Tool Response:**

```
[FILE] config.json
[FILE] pytorch_model.bin
```

**🤖 Agent** Let me check if there are any evaluation results in the main evaluation directory:

**🔧 filesystem directory_tree**

```
"path": "workspace/evaluation"
```

**↰ Tool Response:**

```
[
  {
    "name": ".setup.py.swp",
    "type": "file"
  },
  {
    "name": "benchmarks",
    "type": "directory",
    "children": [
    {
      "name": "code_generation",
      "type": "directory",
      "children": [
      {
        "name": "eval.py",
        "type": "file"
      }
      ]
    },

    ...
  }
]
```

**🤖 Agent**  Let me check the main evaluation script to understand how evaluation results are generated:

**🔧 filesystem read_file**

```
"path": "eval.py"
```

**↰ Tool Response:**

```
...

def run_benchmark_evaluation(benchmark_name, model_path):

...
```

**🤖 Agent**  Now I need to run the evaluation script for each checkpoint to find the one with the highest eval_accuracy. Let me run the evaluation for each checkpoint:

**🔧 terminal run_command**

```
"command": "cd workspace && python evaluation/eval.py checkpoints/step_100"
```

**↩ Tool Response:**

```
[
   {"type": "text", "text": "0.493 ", "annotations": null},
   {"type": "text", "text": "math_reasoning: 0.345 code_generation: Error
text_classification: Error sentiment_analysis: 0.617 question_answering:
0.475 logical_reasoning: 0.319 common_sense: 0.53 reading_comprehension:
0.475 dialogue_generation: Error summarization: 0.517 translation: 0.64
knowledge_retrieval: 0.529 creative_writing: 0.328 instruction_following:
0.55 safety_evaluation: 0.628 ", "annotations": null, "error": true}
   ...
]
```

Processing other model checkpoints ( `Step_200` - `Step_900` ) ......

**🔧 terminal run_command**

```
"command": "cd workspace && python evaluation/eval.py checkpoints/step_1000"
```

**↩ Tool Response:**

```
[
   {"type": "text", "text": "0.712 ", "annotations": null},
   ...
]
```

**🤖 Agent** Perfect! I can see that step_1000 has the highest eval_accuracy of 0.712. Let me also extract the individual benchmark scores for step_1000. I notice there are some errors with code_generation, text_classification, and dialogue_generation benchmarks, but I have scores for the other benchmarks. Let me fix those missing benchmarks first to get complete results. Let me check what's wrong with the code_generation benchmark:

**🔧 filesystem read_file**

```
"path": "workspace/evaluation/benchmarks/code_generation/eval.py"
```

**↩ Tool Response:**

```
...

   if not os.path.isfile(args.model_path):

...
```

**🤖 Agent** I see the issue - the code_generation script is checking if the model_path is a file, but we're passing a directory. Let me check the text_classification script:

**🔧 filesystem read_file**

```
"path": "workspace/evaluation/benchmarks/text_classification/eval.py"
```

**↩ Tool Response:**

```
...

from utils.benchmark_utils import get_benchmark_score

...
```

**🤖 Agent** Let me check the dialogue_generation script:

**🔧 filesystem read_file**

```
"path": "workspace/evaluation/benchmarks/dialogue_generation/eval.py"
```

**↩ Tool Response:**

```
...

from utils.benchmark_utils import get_benchmark_score

...
```

**🤖 Agent** Let me check the benchmark_utils to understand how the scoring works:

**🔧 terminal run_command**

```
"command": "cd workspace/evaluation &&
python  -c  "from  utils.benchmark_utils  import  get_benchmark_score;
print(get_benchmark_score('code_generation', 1000))""
```

**↩ Tool Response:**

```
Security    violation:      Invalid    command    part   'python   -c   "from
utils.benchmark_utils import get_benchmark_score': Invalid command format:
No closing quotation
```

**🔧 python-execute**

```
import sys
import os
sys.path.insert(0, '/workspace/dumps/workspace/evaluation')
from utils.benchmark_utils import get_benchmark_score

# Get scores for step_1000 for the missing benchmarks
code_gen_score = get_benchmark_score("code_generation", 1000)
text_class_score = get_benchmark_score("text_classification", 1000)
dialogue_gen_score = get_benchmark_score("dialogue_generation", 1000)

print(f"code_generation: code_gen_score")
print(f"text_classification: text_class_score")
print(f"dialogue_generation: dialogue_gen_score")
```

**↩ Tool Response:**

```
=== STDOUT ===
code_generation: 0.65
text_classification: 0.828
dialogue_generation: 0.644
```

**🤖 Agent** Great! Now I have all the scores for step_1000. Let me compile the complete results for step_1000:Based on the evaluation results, step_1000 has the highest overall eval_accuracy of 0.712. Here are the complete results for step_1000:

- Math Reasoning: 0.550
- Code Generation: 0.650
- ...

Now I need to:

- Get the HF token
- Push the step_1000 model to Hugging Face Hub
- Update the README.md with these results

Let me first get the HF token:

**🔧 filesystem read_file**

```
"path": "workspace/hf_token.txt"
```

**↩ Tool Response:**

```
hf_xxxxxxxxxxxxxxx
```

**🤖 Agent** Now let me authenticate with Hugging Face:

**🔧 terminal run_command**

```
"command": "cd workspace && echo "hf_xxxxxxxxxxxxxxx" | huggingface-cli login
--token"
```

**↩ Tool Response:**

```
Security violation: Invalid command part 'huggingface-cli login --token':
Command 'huggingface-cli' is not allowed
```

**🤖 Agent** Let me try using the Python API instead:

```
python-execute

import os
from huggingface_hub import HfApi, login

# Set the token
token = "hf_xxxxxxxxxxxxxxx"
os.environ["HF_TOKEN"] = token

# Login
login(token=token)

# Initialize API
api = HfApi()

print("Successfully authenticated with Hugging Face Hub")
```

↩ **Tool Response:**

```
=== STDOUT ===
Successfully authenticated with Hugging Face Hub
=== STDERR ===
Note: Environment variable'HF_TOKEN' is set and is the current active token
independently from the token you've just configured.
```

🤖 **Agent**  Now let me check what files are in the step_1000 checkpoint:

```
filesystem list_directory

"path": "workspace/checkpoints/step_1000"
```

↩ **Tool Response:**

```
[FILE] config.json
[FILE] pytorch_model.bin
```

🤖 **Agent**  Let me also check if there are any other necessary files (like tokenizer files) in the workspace:

```
filesystem search_files

"path": "workspace", "pattern": "tokenizer"
```

↩ **Tool Response:**

```
No matches found
```

🤖 **Agent**  Let me check if there are any vocab files or other model-related files:

```
filesystem search_files

"path": "workspace", "pattern": "vocab"
```

↩ **Tool Response:**

```
No matches found
```

🤖 **Agent** Let me check the config.json to understand the model better:

🔧 **filesystem read_file**

```
"path": "workspace/checkpoints/step_1000/config.json"
```

↩ **Tool Response:**

```
{
   "model_type": "bert",
   "architectures": ["BertModel"]
}
```

🤖 **Agent** Now let me create the repository and push the model. First, let me copy the best model to a temporary directory and add the updated README:

🔧 **filesystem create_directory**

```
"path": "workspace/model_upload"
```

↩ **Tool Response:**

```
Successfully created directory workspace/model_upload
```

🔧 **terminal run_command**

```
"command": "cd workspace && cp checkpoints/step_1000/* model_upload/"
```

↩ **Tool Response:**

```
Command completed with return code: 0
```

🤖 **Agent** Now let me update the README.md with the evaluation results. I need to replace the {RESULT} placeholders with the actual scores:

🔧 **filesystem read_file**

```
"path": "workspace/README.md"
```

**↩ Tool Response:**

```
...

| | Benchmark | Model1 | Model2 | Model1-v2 | MyAwesomeModel |
|---|---|---|---|---|---|
| **Core Reasoning Tasks** | Math Reasoning | 0.510 | 0.535 | 0.521 |
{RESULT} |
...
| **Language Understanding** | Reading Comprehension | 0.671 | 0.685 | 0.690
| {RESULT} |
...
| **Generation Tasks** | Code Generation | 0.615 | 0.631 | 0.640 | {RESULT}
|
...
| **Specialized Capabilities**| Translation | 0.782 | 0.799 | 0.801 |
{RESULT} |
...

...
```

**🤖 Agent** Now let me create the updated README.md with the actual evaluation results:

**🔧 filesystem edit_file**

```
{
  "path": "README.md",
  "edits": [
    "oldText":"| **Core Reasoning Tasks** | Math Reasoning | 0.510 | 0.535 |
0.521 | {RESULT} |",
    "newText":"| **Core Reasoning Tasks** | Math Reasoning | 0.510 | 0.535 |
0.521 | 0.550 |",
...
  ]
}
```

**↩ Tool Response:**

```
| | Benchmark | Model1 | Model2 | Model1-v2 | MyAwesomeModel |
|---|---|---|---|---|---|
-| Core Reasoning Tasks | Math Reasoning | 0.510 | 0.535 | 0.521 | RESULT |
...
+| Core Reasoning Tasks | Math Reasoning | 0.510 | 0.535 | 0.521 | 0.550 |
...
```

**🤖 Agent** Now let me copy the updated README to the model upload directory:

**🔧 terminal run_command**

```
"command": "cd workspace && cp README.md model_upload/"
```

**↩ Tool Response:**

```
Command completed with return code: 0
```

**🤖 Agent** Now let me push the model to Hugging Face Hub:

🔧 **python-execute**

```python
import os
from huggingface_hub import HfApi, login

# Set the token and login
token = "hf_xxxxxxxxxxxxxxxx"
os.environ["HF_TOKEN"] = token
login(token=token)

# Initialize API
api = HfApi()

# Create the repository
repo_id = "MyAwesomeModel-TestRepo"
try:
    api.create_repo(repo_id=repo_id,    private=False, exist_ok=True)
    print(f"Repository repo_id created successfully!")
except Exception as e:
    print(f"Error creating repository: e")

# Upload all files from the model_upload directory
model_path = "/workspace/dumps/workspace/model_upload"
try:
    api.upload_folder(
        folder_path=model_path,
        repo_id=repo_id,
        repo_type="model"
    )
    print(f"Model uploaded successfully to repo_id!")
except Exception as e:
    print(f"Error uploading model: e")
```

↩ **Tool Response:**

```
=== STDOUT ===
Repository MyAwesomeModel-TestRepo created successfully!
Error uploading model: 404 Client Error.

Repository Not Found.
=== STDERR ===
Note: Environment variable‘HF_TOKEN‘ is set and is the current active token
independently from the token you've just configured.
```

🤖 **Agent** I need to include my username in the repository ID. Let me check who I am:

🔧 **huggingface hf_whoami**

```
None
```

↩ **Tool Response:**

```
You are authenticated as anonymized_user.
```

🤖 **Agent** Now let me create the repository with the correct format:

**🔧 python-execute**

```
...
repo_id = "anonymized_user/MyAwesomeModel-TestRepo"
...
```

**↩ Tool Response:**

```
=== STDOUT ===
Repository anonymized_user/MyAwesomeModel-TestRepo created successfully!
Model uploaded successfully to anonymized_user/MyAwesomeModel-TestRepo!
=== STDERR === Note: Environment variable'HF_TOKEN' is set and is the current
active token independently from the token you've just configured.
```

**🤖 Agent** Perfect! Let me verify what files were uploaded:

**🔧 filesystem list_directory**

```
"path": "workspace/model_upload"
```

**↩ Tool Response:**

```
[FILE] README.md
[FILE] config.json
[FILE] pytorch_model.bin
```

**🤖 Agent** Excellent! Let me also verify the README.md content in the upload directory to make sure it has the correct evaluation results:

**🔧 filesystem read_file**

```
{
    "path": "workspace/model_upload/README.md",
    "head": 50
}
```

**↩ Tool Response:**

```
license: mit
library_name: transformers
---
# MyAwesomeModel
...
```

**🤖 Agent** Perfect! The task has been completed successfully. ...

**🔧 claim_done**

```
None
```

**↩ Tool Response:**

```
you have claimed the task is done!
```

**Traj #2: Notion HR (Claude-4-Sonnet, succeed after 45 tool callings)**

👤 **User**  Please update the candidate information on the HR Record subpage of the Notion page according to the resumes in my workspace. All information must be filled out strictly according to the content in the resumes, without making any unauthorized modifications or adding/removing any words. Also, please delete the existing sample entries in the record table. At the same time, if the position applied for by the applicant is currently not open for recruitment, please send an email to the corresponding applicant using the following template information (including line breaks). Do not send the email by mistake:

*Dear Applicant,*

*Thank you very much for your interest in joining MCP Inc and for taking the time to submit your application. We truly appreciate the effort you put into applying for this position.*

*After careful review, we regret to inform you that the position you applied for is currently filled. However, we were impressed with your qualifications and would like to keep your resume on file for future opportunities that may better match your skills and experience.*

*We encourage you to check our careers page regularly for new openings, and please feel free to apply for other positions that align with your background and interests.*

*Thank you again for considering MCP Inc as a potential employer. We wish you the very best in your job search and future career endeavors.*

*Best regards, HR Team, MCP Inc*

🤖 **Agent**  I'll help you update the HR Record subpage in Notion with resume information and handle any necessary email communications. Let me start by exploring your workspace to find the resumes and then locate the Notion page.

🔧 **filesystem list_directory**

```
"path": "workspace"
```

↩ **Tool Response:**

```
[DIR] .pdf_tools_tempfiles
[DIR] cvs
```

🔧 **filesystem list_directory**

```
"path": "workspace/cvs"
```

↩ **Tool Response:**

```
[FILE] alice_jackson_resume.pdf
[FILE] angela_moore_resume.pdf
[FILE] charles_castillo_resume.pdf
...
[FILE] nancy_robinson_resume.pdf
```

🤖 **Agent**  Great! I found 10 resume PDF files. Now let me search for the Notion page with HR Record information.

🔧 **notion API-post-search**

> "query": "HR Record"

↩ **Tool Response:**

> {"object":"list","results":[...]}

🤖 **Agent** I found the HR Record page. Let me retrieve its contents to see the structure and any existing database.

🔧 **notion API-get-block-children**

> "block_id": "xxxxx"

↩ **Tool Response:**

> {"object":"list","results":[...]}

🤖 **Agent** I can see there are three databases in the HR Record page:

1. Job Positions (block ID: xx-yy-zz)
2. Job Application Timeline (block ID: yy-zz-xx)
3. Candidates (block ID: zz-xx-yy)

Let me first examine the Job Positions database to understand what positions are available, and then the Candidates database to see its structure.

🔧 **notion API-retrieve-a-database**

> "database_id": "xx-yy-zz"

↩ **Tool Response:**

> {"object":"database","id":"xx-yy-zz",...}

🔧 **notion API-post-database-query**

> "database_id": "xx-yy-zz"

↩ **Tool Response:**

> {"object":"list","results":[...]}

🤖 **Agent** Now let me examine the Candidates database structure:

🔧 **notion API-retrieve-a-database**

> "database_id": "zz-xx-yy"

**↩ Tool Response:**

{"object":"database","id":"zz-xx-yy",...}

**🔧 notion API-post-database-query**

"database_id": "zz-xx-yy"

**↩ Tool Response:**

{"object":"list","results":[...]}

**🤖 Agent** Now I have a clear understanding of the situation:Job Positions with available headcount (Head Counts > 0):

- Advanced Investment Banking Analyst: 3 positions
- Market Risk Specialist: 4 positions
- ...

Existing sample candidates and their applied positions:

- Alice Jackson → Financial Risk Analyst (2 positions available) ✓
- Debra Smith → Securities Trader (0 positions available) ✗
- Michael Diaz → Investment Analyst (2 positions available) ✓
- ...

I need to:

1. Delete all existing sample entries
2. Read all the resume PDFs to extract candidate information
3. Add new entries based on actual resumes
4. Send rejection emails to candidates applying for positions with 0 headcount

Let me start by reading all the resume PDFs:

**🔧 pdf-tools read_pdf_pages**

"pdf_file_path": "workspace/cvs/alice_jackson_resume.pdf", "start_page": 1, "end_page": 10

**↩ Tool Response:**

```
PDF total pages: 3
Extracted page range: 1-3

=== Page 1 ===
ALICE JACKSON
**Applied Position: Financial Risk Analyst **
CONTACT INFORMATION
...
```

Read the resumes of the other candidates in sequence ......

🤖 **Agent**  Now I have all the resume information. Let me analyze what I found:
**Candidates from resumes and their applied positions:**

- **Alice Jackson** – Applied: Financial Risk Analyst (2 positions available) ✓
  Education: Master of Finance, Columbia Business School; Bachelor Economics, University of Pennsylvania
  Email: alice_jackson38@mcp.com

- **Angela Moore** – Applied: Securities Trader (0 positions available) ✗
  Education: Master of Science in Computational Finance, Carnegie Mellon University
  Email: angela_moore89@mcp.com

- ...

Now I need to:

1. Delete all existing sample candidates

2. Add the real candidates from resumes

3. Send rejection emails to candidates applying for positions with 0 headcount

Let me start by deleting the existing sample candidates:

🔧 **notion API-update-a-block**

```
"block_id": "xy-yz-zx", "archived": true
```

↩ **Tool Response:**

```
"status":400,"object":"error","code":"validation_error",...
```

🤖 **Agent**  I need to use the correct API to delete pages from the database. Let me delete them one by one:

🔧 **notion API-patch-page**

```
"page_id": "xy-yz-zx", "archived": true
```

↩ **Tool Response:**

```
{"object":"page","id":"xy-yz-zx",...}
```

Delete others one by one ......

🤖 **Agent**  Great! I've deleted all the existing sample candidates. Now I need to create new pages for the actual candidates based on their resumes. Let me start creating them one by one:

🔧 **notion API-post-page**

```
"parent": {"database_id":"zz-xx-yy"},
"properties": {
  "title":[{"text":{"content":"Alice Jackson"}}], "type":"title"
}
```

↩ **Tool Response:**

```
{"status":400,"object":"error","code":"validation_error",...}
```

🤖 **Agent** I need to use the correct format for creating pages in a database. Let me check the database structure again and create the pages properly:

🔧 `notion API-post-page`

```
{
  "parent": {"database_id":"zz-xx-yy"},
  "properties": {"Name":{"title":[{"text":{"content":"Alice Jackson"}}]},
  "Email":{"rich_text":[{"text":{"content":"alice_jackson38@mcp.com"}}]},
  "Applied    Position":{"rich_text":[{"text":{"content":"Financial    Risk
Analyst"}}]},
  "School":{"rich_text":[{"text":{"content":"Columbia Business School"}}]},
  "Highest Degree":{"select":{"name":"master"}}}
}
```

↩ **Tool Response:**

```
{"object":"page","id":"yx-zy-xz",...}
```

🤖 **Agent** Great! Now let me add the remaining candidates:

Add the remaining candidates ......

🤖 **Agent** Perfect! I've successfully added all 10 candidates from the resumes to the Notion database. Now I need to send rejection emails to candidates who applied for positions that are currently not open for recruitment (positions with 0 headcount).Based on my analysis, the candidates who need rejection emails are:

- Angela Moore (Securities Trader - 0 positions)
- Debra Smith (Securities Trader - 0 positions)
- Emily James (Securities Trader - 0 positions)
- Martha Morales (Portfolio Manager - 0 positions)

Let me send the rejection emails:

🔧 `emails send_email`

```
"to": "angela_moore89@mcp.com",
"subject": "Application Status Update - Securities Trader Position",
"body":...
```

↩ **Tool Response:**

```
Email sent successfully to angela_moore89@mcp.com
```

Send other rejection emails ......

🤖 **Agent** Excellent! I have successfully completed all the required tasks: ...

🔧 `claim_done`

```
None
```

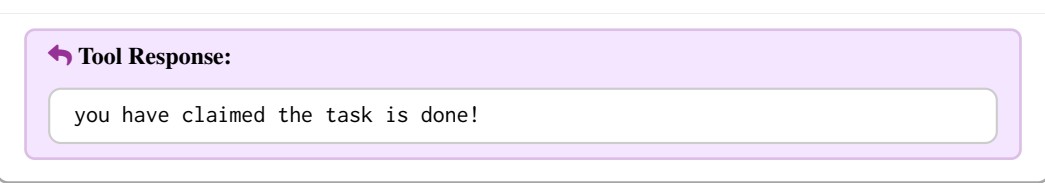

↩ **Tool Response:**

```
you have claimed the task is done!
```

