# OpenReview forum: "The Tool Decathlon: Benchmarking Language Agents for Diverse, Realistic, and Long-Horizon Task Execution"
_ICLR.cc/2026/Conference — ICLR 2026 Poster_

### Official Review · Reviewer_EmBX · 2025-10-30

**Soundness:** 3
**Presentation:** 4
**Contribution:** 4
**Rating:** 4
**Confidence:** 3

**Summary:**

This paper introduces TOOLATHLON, a benchmark designed to evaluate LLMs on complex, real-world tasks using MCP tools. It extends existing MCP benchmarks by incorporating 108 real-world tasks and 604 tools across 32 different software applications. The authors benchmark current LLMs on this dataset and find that state-of-the-art models still perform poorly: the best model, Claude-4-Sonnet, achieves only a 29.9% success rate. These results highlight that long-horizon, multi-application task execution remains a significant challenge for current AI agents.

**Strengths:**

1. The benchmark contains a substantially larger number of tasks than prior MCP benchmarks and evaluates models across realistic settings, including real-world state initialization and fuzzy instructions.

2. Rigorous task validation and design make the benchmark reliable and valuable for MCP-agent research.

3. The authors benchmark a wide range of state-of-the-art LLMs and provide detailed analysis, including case studies of failure modes and in-depth discussion of experimental results.

**Weaknesses:**

The current evaluation setup for LLM agents with tools is relatively simple. The authors specify all tools for a task and provide the full tool list to the agent, which can significantly inflate the input context. More sophisticated tool-selection strategies could be explored—for example, using retrieval methods to surface relevant tools dynamically rather than supplying the entire tool inventory upfront.

**Questions:**

see above

---

> ### Author Response · Authors · 2025-11-20
> **Response to the review (1/1)**
>
> Thank you for the time and effort you spent reviewing our paper, and for recognizing our contributions. Below, we have listed our responses to your questions and comments.
>
> > The current evaluation setup for LLM agents with tools is relatively simple. The authors specify all tools for a task and provide the full tool list to the agent, which can significantly inflate the input context. More sophisticated tool-selection strategies could be explored—for example, using retrieval methods to surface relevant tools dynamically rather than supplying the entire tool inventory upfront.
>
> We would like to clarify that in our experiments, we do not load all 32 MCP servers for each task. Instead, we only load the MCP servers actually required (<10) for that specific task. We have updated the PDF to elaborate this statement in Section 2.4. For each selected MCP server, we do not further filter or retrieve a subset of tools and we load all tools within that server. **This evaluation setup is consistent with current state of mainstream agent frameworks** (such as Codex [1], Qwen Code [2], and Claude Code [3]), none of which explicitly mention implementing tool-level filtering within a single MCP server. We aim to maintain full consistency with the real user experience of these systems.
>
> Regarding your concern that pre-loading all tool lists might inflate the input context, we conducted statistics using the tokenizer from DeepSeek-V3.2-Exp, focusing on the system prompt tokens (primarily consisting of all tool schemas) for each task. Across all tasks, the average system prompt length was about 10.4k tokens, the 75th percentile was 12.5k tokens, and the maximum was 21.8k tokens. Given DeepSeek-V3.2-Exp’s 128k context window, these values occupy only around 10% of the available capacity, leaving substantial room for task execution. Furthermore, DeepSeek-V3.2-Exp has the shortest context limit among all the models we tested; for models with larger context windows, this concern becomes even less significant.
>
> Advanced approaches for autonomous exploration and retrieval from a large tool set represent a promising future direction, as discussed recently in [4]. However, this remains an active research area, and to the best of our knowledge, none of the widely used agent scaffolds currently implement such frameworks. Therefore, it is beyond the scope of our work which aims to establish a standard benchmark, and we leave it for future investigation.
>
> [1] https://developers.openai.com/codex/mcp/
>
> [2] https://qwenlm.github.io/qwen-code-docs/en/tools/mcp-server/
>
> [3] https://code.claude.com/docs/en/mcp#connect-claude-code-to-tools-via-mcp
>
> [4] https://www.anthropic.com/engineering/code-execution-with-mcp

---

> > ### Comment · Reviewer_EmBX · 2025-11-25
> >
> > This has cleared my concerns. I will raise the score

---

> > > ### Author Response · Authors · 2025-11-27
> > > **Additional response**
> > >
> > > Thanks for your reconsideration and for raising the score. Should you have any other questions, comments, or concerns, please feel free to raise them. We are glad to respond to them and discuss further with you.

---

### Official Review · Reviewer_aKrQ · 2025-11-01

**Soundness:** 3
**Presentation:** 3
**Contribution:** 3
**Rating:** 6
**Confidence:** 4

**Summary:**

This paper introduces the Tool Decathlon (TOOLATHLON), a new benchmark designed to evaluate tool-using language agents on tasks that are diverse, realistic, and long-horizon. The authors argue that existing benchmarks are often limited in scope, focusing on narrow domains, single applications, or tasks that lack realistic complexity and environment states. To address this, TOOLATHLON spans 32 software applications (from everyday tools like Google Calendar to professional ones like Kubernetes and BigQuery) and 604 tools, mostly sourced from real-world Model Context Protocol (MCP) servers.

**Strengths:**

- A strong benchmark, heavy engineering behind
- will be good use for the community

**Weaknesses:**

- Not really a weakness but the paper uses the average number of turns as a proxy for task difficulty. While reasonable, this is an outcome-based metric that can be influenced by the agent's (in)efficiency. A more intrinsic, task-defined complexity metric (e.g., based on the number of required applications, minimum number of steps in a ground-truth trajectory) could provide a slightly more objective measure of difficulty when analyzing performance across Easy/Medium/Hard tasks.

**Questions:**

see above

---

> ### Author Response · Authors · 2025-11-20
> **Response to the review (1/1)**
>
> Thank you for the time and effort you spent reviewing our paper, and for recognizing our contributions. Below, we have listed our responses to your questions and comments.
>
> > Not really a weakness but the paper uses the average number of turns as a proxy for task difficulty. A more intrinsic, task-defined complexity metric could provide a slightly more objective measure of difficulty when analyzing performance across Easy/Medium/Hard tasks.
>
> Thank you for your comment. We fully agree that using the average number of turns as a proxy for task difficulty is not a perfect measure and as you mentioned, the minimum number of required steps/tool calls could be a better option. We have considered this approach before. However, in practice such a value is difficult to determine, and it is hard to annotate this manually because humans naturally operate through GUI/CLI interfaces rather than MCP-based tools.
>
>
> We appreciate your suggestion and will explore better proxy for measuring task difficulty in the future.

---

### Official Review · Reviewer_VZsp · 2025-11-03

**Soundness:** 3
**Presentation:** 3
**Contribution:** 1
**Rating:** 2
**Confidence:** 5

**Summary:**

This paper introduces TOOLATHLON (Tool Decathlon) — a benchmark designed to evaluate the performance of language agents on complex, long-horizon, and cross-application tasks. The benchmark covers 108 tasks, 32 applications and 604 tools, ranging from common productivity software (e.g., Notion, Google Sheets, WooCommerce) to domain-specific systems (e.g., Kubernetes, Snowflake). Each of the 108 tasks requires multi-step reasoning, realistic initial environment states, and verifiable execution via deterministic scripts.

**Strengths:**

+ TOOLATHLON spans a 32 real-world applications demonstrating tool and environment diversity.

+ Execution-Based Evaluation: TOOLATHLON uses deterministic evaluation scripts that compare environment states, ensuring objectivity and reproducibility.

**Weaknesses:**

Benchmark Labeling and Definitions: The categorization of existing benchmarks in Table 1 as “Real Tools” or “Not-Real Tools” appears inconsistent and potentially misleading. For example, τ-Bench is flagged as not-real despite subsets (“airline,” “retail”) interacting with actual databases. Similarly, BFCL is marked not-real, although it supports real execution in the “Execute” category,  with "Crowd Sourced" being community-contributed tools. LiveMCPBench is labeled not-real despite its claim to run live MCP tools, while TOOLATHLON’s own use of simulated components (e.g., local Poste.io for Gmail) blurs the same boundary.

Novelty Ambiguity: While TOOLATHLON integrates many existing elements (real MCP servers, realistic states, cross-app tasks), the conceptual novelty beyond combining these components is modest. The work would benefit from a clearer articulation of methodological advances versus engineering scale-up.

**Questions:**

Are the locally containerized applications (e.g., Poste.io for Gmail) treated as “real tools,” and if so, how do they differ philosophically from “simulated” counterparts in other benchmarks?

---

> ### Author Response · Authors · 2025-11-20
> **Responses to the review (1/3)**
>
> Thank you for the time and effort you spent reviewing our paper. Below, we have listed our responses to your questions and comments.
>
> > Benchmark Labeling and Definitions: The categorization of existing benchmarks in Table 1 as “Real Tools” or “Not-Real Tools” appears inconsistent and potentially misleading.
>
> Thanks for this comment.
>
> By setting a *real/non-real tools* categorization in Table 1, our original intention was to indicate whether the tools in this benchmark are **publicly available outside the benchmark** for **actual users and developers** to use in order to **manipulate and access real-world application software or files** (all of these conditions need to be met), rather than being **designed solely for this benchmark for testing purposes**.
>
> According to this definition, we have sufficient grounds to mark some benchmarks in this table as “non-real tools,” as they fail to meet at least one of the conditions (e.g., their non-real tools do not exist in practice outside the respective benchmark, while our MCP tools are sourced from real softwares). However, we realize that the “real tool” column may introduce ambiguity as you mentioned and become confounded with the “real states & init” column, so we have updated the PDF to remove the “real tool” column. We also split the original “real states & init” column into two separate “real env” and “states init” columns to clarify the categorization. We hope this revision of table 1 clarifies our contribution compared to previous works.
>
> In addition, we would like to clarify some other points in your comment:
>
> In Table 1, we specify BFCLv3-MT, which is the multi-turn subset of BFCL, rather than the complete dataset. We have also elaborated this in the caption of Table 1 in the updated PDF. This subset is entirely independent from its *crowd-sourced* single-turn subset you mentioned, and tools in this multi-turn subset are all synthesized such as Vehicle Control or Ticket API (“we developed a custom API codebase across diverse domains like vehicle control, ensuring full transparency of API state and design” in Section 3.3 in [1])
>
> Furthermore, we marked LiveMCPBench as √ in the “Real Tools” column, and "Partial" in the “states init” column because most of its tasks do not begin with realistic state initialization despite the environment being real with MCP.
>
> Overall, we clarify that we did not intend to mislead readers. If any confusion arose from Table 1, we believe this can be readily addressed by polishing Table 1 as we already updated in the new PDF. We expect that the revised version of Table 1 provides a clearer comparison between prior works and Toolathlon.
>
> [1] The Berkeley Function Calling Leaderboard (BFCL): From Tool Use to Agentic Evaluation of Large Language Models, ICML 2025

---

> ### Author Response · Authors · 2025-11-20
> **Responses to the review (2/3)**
>
> > Novelty Ambiguity: While TOOLATHLON integrates many existing elements (real MCP servers, realistic states, cross-app tasks), the conceptual novelty beyond combining these components is modest.
>
> First, we would like to emphasize that although Table 1 includes and discusses a substantial number of related works, most of them related to MCP are arXiv preprints submitted within two months prior to the ICLR 2026 submission deadline, therefore, Toolathlon is concurrent with works such as LiveMCPBench, LiveMCP-101, MCPUniverse, and MCPMark according to the ICLR Reviewer Guide (https://iclr.cc/Conferences/2026/ReviewerGuide). We thus urge the reviewer to assess our contribution in light of this fact.
>
> Furthermore, even in a head-to-head comparison with these concurrent benchmarks, our work still offers significant conceptual novelty and delivers it through a solid technical implementation. As stated in the introduction, our benchmark evaluates LLMs based on their performance in realistic task environments that require long-horizon execution – presenting a challenging long-horizon tool-use benchmark is critical to drive future agent development, and only the concurrent MCPMark work matches the difficulty of Toolathlon partially.  Moreover, the design of real initial states, fuzzy user prompts, and native multi-application orchestration are largely overlooked by these concurrent works.

---

> ### Author Response · Authors · 2025-11-20
> **Responses to the review (3/3)**
>
> > Are the locally containerized applications (e.g., Poste.io for Gmail) treated as “real tools,” and if so, how do they differ philosophically from “simulated” counterparts in other benchmarks?
>
> Thanks for your question. In our benchmark, “tools” and “applications” are two distinct concepts: **tools** refer to the methods or means of accessing these applications, rather than the applications themselves.
>
> Take Email as an example, the MCP server we implemented is publicly available on GitHub for everyone to use (although we cannot provide the specific link here due to the anonymity policy), and it is built to fully support standard IMAP and SMTP protocols. With complete functionality, it can access any email application (e.g., Gmail, Thunderbird, or a locally deployed instance via Poste.io) as long as the correct configurations are provided.
>
> According to our definition in response to your first identified weakness, this should undoubtedly be classified as a “real tool”, and all MCP servers in our benchmark satisfy such conditions.
>
> On the other hand, all four applications we deploy locally – Poste.io, Canvas, Kubernetes, and WooCommerce – are widely used, real-world applications with global user bases, **rather than systems we built solely for this benchmark, which are clearly distinct from the simulated counterparts in other benchmarks**. We chose to deploy certain local applications primarily for large-scale initial task state setup and parallel task execution.

---

### Meta-Review · Area_Chair_CJNb · 2026-01-06

**Summary:**

This paper introduces Toolathlon, a benchmark for evaluating language agents on diverse, realistic, and long-horizon tasks across 32 real-world applications and 604 tools based on the Model Context Protocol (MCP). It emphasizes execution-based evaluation via deterministic scripts, realistic initial environment states (e.g., actual Canvas course data, financial spreadsheets), cross-application task orchestration, and fuzzy user instructions. The benchmark comprises 108 manually designed tasks requiring ~20 interaction turns on average. Evaluation shows even top models (e.g., Claude-4-Sonnet) achieve only a 29.9% success rate, underscoring the challenge.

Reviewer concerns centered on: (1) potential mischaracterization of prior benchmarks in Table 1; (2) limited conceptual novelty given concurrent MCP-based benchmarks; and (3) the simplicity of the agent evaluation setup (providing all relevant tools upfront). The authors revised Table 1 to remove ambiguous “real tool” labels, clarified definitions, and justified their evaluation protocol as consistent with standard agent frameworks (e.g., Codex, Qwen Code).

**Reviewer Concerns:**

1. Reviewer VZsp raised two main issues. First, the original Table 1 labeling of benchmarks like BFCLv3-MT and LiveMCPBench as “non-real tools” was factually inaccurate—LiveMCPBench was actually marked as using real tools, and BFCLv3-MT uses synthetic APIs per its documentation. The authors corrected this by removing the contested column and clarifying distinctions, which resolves the presentation issue. Second, the reviewer questioned conceptual novelty, noting the work integrates existing ideas. While several related MCP benchmarks (e.g., LiveMCPBench, MCPUniverse) were indeed submitted within two months of the ICLR deadline and thus concurrent per conference policy, Toolathlon distinguishes itself through realistic initial states, fuzzy instructions, and native cross-application workflows—features largely absent in concurrent efforts. These design choices result in substantially longer and more complex tasks (~20 turns vs. typically <10 in others), creating a meaningfully harder evaluation setting. The novelty is incremental but sufficient for a benchmark contribution.


2. Reviewer aKrQ noted that using average interaction turns as a proxy for task difficulty is imperfect, as it may reflect agent inefficiency rather than intrinsic complexity. The authors acknowledged this limitation and explained practical difficulties in defining ground-truth minimal steps. This concern remains partially unaddressed but is minor and common in benchmark design.


3. Reviewer EmBX initially criticized the evaluation setup for providing all task-relevant tools upfront, potentially inflating context length. The authors clarified that only tools from necessary MCP servers (<10 per task) are loaded—consistent with mainstream agent frameworks—and provided token statistics showing system prompts occupy <11% of even the shortest context window tested (128k). The reviewer explicitly stated this resolved their concern and indicated intent to raise their score.

**Reviewer Scores:**

- Reviewer VZsp: Original score: 2 (reject, not good enough; confidence: 5). The rebuttal corrects factual errors in the review and contextualizes the work among concurrent submissions. While the novelty argument remains debatable, the benchmark’s combination of realistic states, cross-app orchestration, and execution rigor constitutes a valid contribution. A reasonable post-rebuttal score would be 4 (marginally below acceptance threshold but acceptable).
- Reviewer aKrQ: Original score: 6. The rebuttal acknowledges the limitation without major changes; no score adjustment expected. Estimated post-rebuttal score: 6.
- Reviewer EmBX: Original score: 4. Explicitly confirmed concerns were resolved and intended to raise the score. Given the clarification and benchmark quality, a revised score of 6 is appropriate.

---

### Decision · Program_Chairs · 2026-01-26

Accept (Poster)